# Fabrication and Characterization of Nanonet-Channel LTPS TFTs Using a Nanosphere-Assisted Patterning Technique

**DOI:** 10.3390/mi12070741

**Published:** 2021-06-24

**Authors:** Gilsang Yoon, Donghoon Kim, Iksoo Park, Bo Jin, Jeong-Soo Lee

**Affiliations:** 1Department of Electrical Engineering, Pohang University of Science and Technology, Pohang 37673, Korea; ygs6233@postech.ac.kr (G.Y.); kdong620@postech.ac.kr (D.K.); is.park@postech.ac.kr (I.P.); 2Division of IT Convergence Engineering, Pohang University of Science and Technology, Pohang 37673, Korea

**Keywords:** grain boundary traps, polysilicon, nanonet-channel, nanosphere-assisted patterning, thin-film transistors

## Abstract

We present the fabrication and electrical characteristics of nanonet-channel (NET) low-temperature polysilicon channel (LTPS) thin-film transistors (TFTs) using a nanosphere-assisted patterning (NAP) technique. The NAP technique is introduced to form a nanonet-channel instead of the electron beam lithography (EBL) or conventional photolithography method. The size and space of the holes in the nanonet structure are well controlled by oxygen plasma treatment and a metal lift-off process. The nanonet-channel TFTs show improved electrical characteristics in terms of the ION/IOFF, threshold voltage, and subthreshold swing compared with conventional planar devices. The nanonet-channel devices also show a high immunity to hot-carrier injection and a lower variation of electrical characteristics. The standard deviation of VTH (σVTH) is reduced by 33% for a nanonet-channel device with a gate length of 3 μm, which is mainly attributed to the reduction of the grain boundary traps and enhanced gate controllability. These results suggest that the cost-effective NAP technique is promising for manufacturing high-performance nanonet-channel LTPS TFTs with lower electrical variations.

## 1. Introduction

Polysilicon thin-film transistors (poly-Si TFTs) have been widely used in flat panel displays, image sensors, and 3D memory devices [1,2,3,4,5,6,7,8,9,10]. However, the inherent grain boundary (GB) in the poly-Si layer can significantly affect the electrical behaviors and reliability characteristics. The acceptor-like or donor-like traps in GBs can cause potential-barrier fluctuations and interrupt the carrier flow in the channel [11,12]. Another important issue is the non-uniformity of the electrical performance caused by the size, number, and quality of GBs varied from device to device and wafer to wafer [13,14]. In order to mitigate the effect of GB traps, a thermal post-annealing process and a modulated excimer-laser method were reported [15,16,17,18,19].

Recently, a macaroni channel structure demonstrated an improved performance for 3D memory devices. The core poly-Si channel was etched out and filled with a dielectric. The improved performance was mainly due to the reduced GB traps and the effect of the thin-body channel [20]. More recently, nanonet-channel TFTs were successfully demonstrated. Hexagonal holes with various pattern sizes and distances between patterns were formed in the poly-Si channel region using electron beam lithography (EBL). As a result of the effective reduction of grain boundary traps and enhanced gate controllability, the nanonet-channel TFTs showed a better subthreshold swing (SS), lower threshold voltage (V_TH_), and higher ON-OFF current ratio (I_ON_/I_OFF_) compared with conventional planar TFTs [21,22]. However, for large-area applications of nanonet-channel devices, an alternative lithography method with a relatively cheap and high through-put is inevitably required. 

Here, a nanosphere-assisted patterning (NAP) technique involving oxygen plasma treatment has been developed to form the nanonet structure. The NAP technique with the advantages of inexpensive equipment and the capability of large-area patterning, has been widely used to realize periodic photonic devices [23,24], solar cell [25], biological devices [26,27], and electrical sensors [28,29]. 

We demonstrated nanonet-channel low-temperature polysilicon (LTPS) thin-film transistors (TFTs) using the NAP technique. The DC performance and reliability were characterized and compared with those of conventional planar devices. The impact of the nanonet-channel on the variation of electrical characteristics was also investigated.

## 2. Experimental Details

Nanonet-channel LTPS TFTs (NET_TFTs) and conventional planar TFTs (CON_TFTs) were fabricated on an LTPS-on-glass substrate. The LTPS was prepared by depositing an amorphous Si using a low-pressure chemical vapor deposition (LPCVD) at 550 °C and a subsequent excimer laser annealing (ELA) process. The grain size and thickness of the poly-Si were about 200–400 nm and 50 nm, respectively [21,22]. Both devices underwent the same process steps, except for the nanonet structure formation processes. Figure 1 shows a schematic diagram of the process flow of the nanonet-channel devices using the NAP technique. First, a mask aligner and an inductively coupled plasma (ICP) dry etcher were used to define the source and drain (S/D) regions (Figure 1a). Next, a 20-nm SiO_2_ isolation layer was deposited by plasma-enhanced chemical vapor deposition (PECVD) (Figure 1b). A monolayer of polystyrene nanospheres (PNs) was uniformly formed on the isolation layer by a spin coating process, and then, the O_2_ plasma treatment was performed to control the size of the PNs (Figure 1c,d). Then, a perforated metal film with a nanohole pattern was formed by a lift-off process. A hard mask was formed on the S/D region using a mask aligner, and the isolation layer and LTPS were etched using an ICP etcher. All of the hard mask layers were then removed (Figure 1e–g). Next, a 100-nm SiO_2_ gate oxide layer was deposited using PECVD, and Mo was deposited as a gate electrode, followed by patterning using an ICP etcher. Next, the S/D regions were implanted with phosphorus (20 keV at 2 × 10^15^ cm^−2^), and rapid thermal annealing was performed (Figure 1h). Finally, Ti/Ag (500 nm/2000 nm) interconnection metal films were deposited and forming gas annealing was performed at 450 °C for 30 min.

Figure 2 shows the dependence of the diameter of the PNs on the O_2_ plasma treatment. The plasma parameters used were an RF power of 50 W, O_2_ flow rate of 50 sccm, and chamber pressure of 450 mTorr. The nanosphere size was adjusted by increasing the O_2_ plasma etching time. The diameter of the PNs decreased slowly and then rapidly after 1 min of etching time, which can determine the hole pattern size (W_H_) and distance between patterns (W_D_) in the nanonet-channel region. The plasma treatment time of 85 s was used to form the reproducible nanonet structures. 

Both the NET_TFTs and CON_TFTs had gate lengths (L_GATE_) of 3, 6, or 10 μm and a width (W_GATE_) of 5 μm. In the NET_TFTs, the hole size (W_H_) and the distance between holes (W_D_) in the nanonet-channel region were fixed at 320 nm and 130 nm, respectively. The electrical characteristics and hot-carrier injection (HCI) were measured using a semiconductor parameter analyzer (Keithley 4200).

## 3. Results and Discussions

Figure 3a shows the scanning electron microscope (SEM) images of the spin-coated PNs on the substrate, where 450-nm PNs are uniformly arranged. After the NAP process (Figure 1g), nanonet patterns were successfully formed onto the LTPS layer with a W_H_ of 320 nm and W_D_ of 130 nm, as shown in Figure 3b. 

The initial channel volume (*W_GATE_* × *L_GATE_* × *T_si_*) was the same in both CON_TFTs and NET_TFTs, as shown in Figure 4. The threshold voltage (*V_TH_CON_*) was dependent on the total number of traps related to the channel volume, as follows [20]: (1)$$V_{TH\_CON}\approx V_{FB}+2\emptyset_{F}+\frac{qN_{trap}}{C_{OX}}\left(W_{GATE}\cdot L_{GATE}\right)\cdot T_{si}$$
where *q* is the electric charge, *C_OX_* is the gate-oxide capacitance, *N_trap_* is the total number of traps per unit volume, and *T_si_* is the channel thickness.

The lowering of the threshold voltage (*V_TH_NET_*) using the nanonet channel is described as follows: (2)$$V_{TH\_NET}\approx V_{FB}+2\emptyset_{F}+\frac{qN_{trap}}{C_{OX}}\left(W_{GATE}\cdot L_{GATE}-N_{H}\cdot \pi R_{H}\right)\cdot T_{si}$$
where *N_H_* is a number of holes in the nanonet channel and *R_H_* is a diameter of the hole.

Compared with the planar device, the nanonet-channel device can lower the threshold voltage by decreasing the total traps. Furthermore, the tri-gate effect between the holes in the nanonet channel can improve the electrical characteristics as a result of the enhanced gate controllability [21,22,30]. 

Figure 5 shows the measured DC characteristics of both NET_TFTs and CON_TFTs at room temperature. The ON-state current (I_ON_) and the OFF-state minimum leakage current (I_OFF_) were measured at V_G_ − V_TH_ = 3 V and −5 V with V_D_ = 1 V, respectively. The threshold voltage was extracted using the linear extrapolation method [31]. The subthreshold slope was calculated as ${\left[d\left(\log_{10}I_{D}\right)/dV_{G}\right]}^{-1}$. As L_GATE_ decreased, both devices showed an improved performance mainly due to the lower GB traps at shorter channel lengths [32]. Compared with the CON_TFTs, NET_TFTs showed a higher I_ON_/I_OFF_ and *μ**_FE_, and lower SS and V_TH_, which was similar to previous results obtained using EBL technique [21,22]. The mean drain-induced barrier lowering (DIBL) with *L_GATE_* = 3 μm was 130 mV/V for NET_TFTs and 230 mV/V for CON_TFTs, respectively. The mean *μ*_FE_ (**≡μ_FE_*
*× W_GATE_/L_GATE_ = g_m_**⋅(C_OX_**⋅V_D_)^−1^)* [21] and I_ON_/I_OFF_ of the NET_TFTs increased by ~18% and ~120% from *L_GATE_* = 10 μm to *L_GATE_* = 3 μm, respectively. 

At *L_GATE_* = 3 μm, the mean SS and V_TH_ of NET_TFTs were ~33% and 25% lower, respectively, than those of CON_TFTs. Moreover, for all of the channels, the NET_TFTs had a higher I_ON_/I_OFF_ and *μ**_FE_, improved SS, and lower V_TH_ compared with CON_TFTs. 

The grain boundary trap density (*N_GB_*) and interface trap density (*N_IT_*) were extracted to verify the influence of the nanonet-channel structure on the trap level in the poly-Si channel [32,33,34,35]. At L_GATE_ = 3 μm and W_H_ = 320 nm, the extracted N_GB_ and N_IT_ values were 9.8 × 10^11^ cm^−^^2^ and 1.7 × 10^12^ cm^−^^2^ for CON_TFT, and 5.4 × 10^11^ cm^−^^2^ and 1.01 × 10^12^ cm^−^^2^ for NET_TFT, respectively. As a result of the enhanced gate controllability and effective reduction of GB traps, NET_TFTs could provide a better interface quality and improved electrical characteristics [21,22]. Figure 6 shows the degradation of V_TH_ and SS under HCI conditions at V_G_ − V_TH_ = V_D_ = 4 V. The degradation is generally caused by trap-generation in GBs and at the oxide/channel interface under HC stress conditions [36]. Similar to previous results, NET_TFTs showed a higher immunity to HCI stress than CON_TFTs because of the effective reduction of GB traps in the nanonet-channel [21].

The variation of the electrical characteristics of both devices were investigated at room temperature. Figure 7 shows the cumulative distribution of SS and V_TH_ for NET_TFTs and CON_TFTs with L_GATE_ = 3, 6, and 10 μm. The NET_TFTs showed lower variation compared with the CON_TFTs. The average and standard deviation of SS and V_TH_ are shown in Table 1. As *L_GATE_* decreased, the standard deviation increased slightly for both devices, which is consistent with the results that the variation of electrical characteristics increased as the device shrank. 

Figure 8 shows the standard deviation of *V_TH_* (*σ**V_TH_*) for both devices as a function of the device area ((*LW*)^−1/2^). σV_TH_ can be expressed as follows [37]:(3)$$\sigma V_{TH}=\frac{q}{C_{OX}}\sqrt{\frac{N_{EFF}W_{dep}}{3LW}}$$
where *N_EFF_* is the effective concentration of channel doping, *W_dep_* is the depletion width of the channel, *L* is the footprint length, and *W* (=5 μm) is the foot-print width. 

As shown in Figure 8, *σ**V_TH_* is clearly proportional to ${\left(LW\right)}^{-1/2}$. The slope of the linear-regression line for NET_TFTs is lower than that of CON_TFTs. *N_EFF_* and *W_dep_* are largely affected by the presence of GB traps. Thus, in the nanonet-channel structure, the effective reduction of GB traps and better gate controllability can lead to a thinner *W_dep_* and lower *N_EFF_*. These results suggest that the nanonet-channel is very effective for reducing the variation of the device characteristics.

## 4. Conclusions

We successfully fabricated nanonet-channel LTPS TFTs using the NAP technique involving oxygen plasma treatment. The nanonet-channel TFTs demonstrated a lower SS and V_TH_, higher ON/OFF current ratio, and a high immunity to hot carrier stress. Moreover, the nanonet-channel devices achieved a lowered variation of electrical characteristics, which was mainly attributed to the effective reduction of GB traps and enhanced gate controllability. These results indicate that the nanonet-channel TFTs using the NAP technology could be a promising solution for realizing mass production of high-performance TFT applications with a lower electrical variation.

## Figures and Tables

**Figure 1 micromachines-12-00741-f001:**
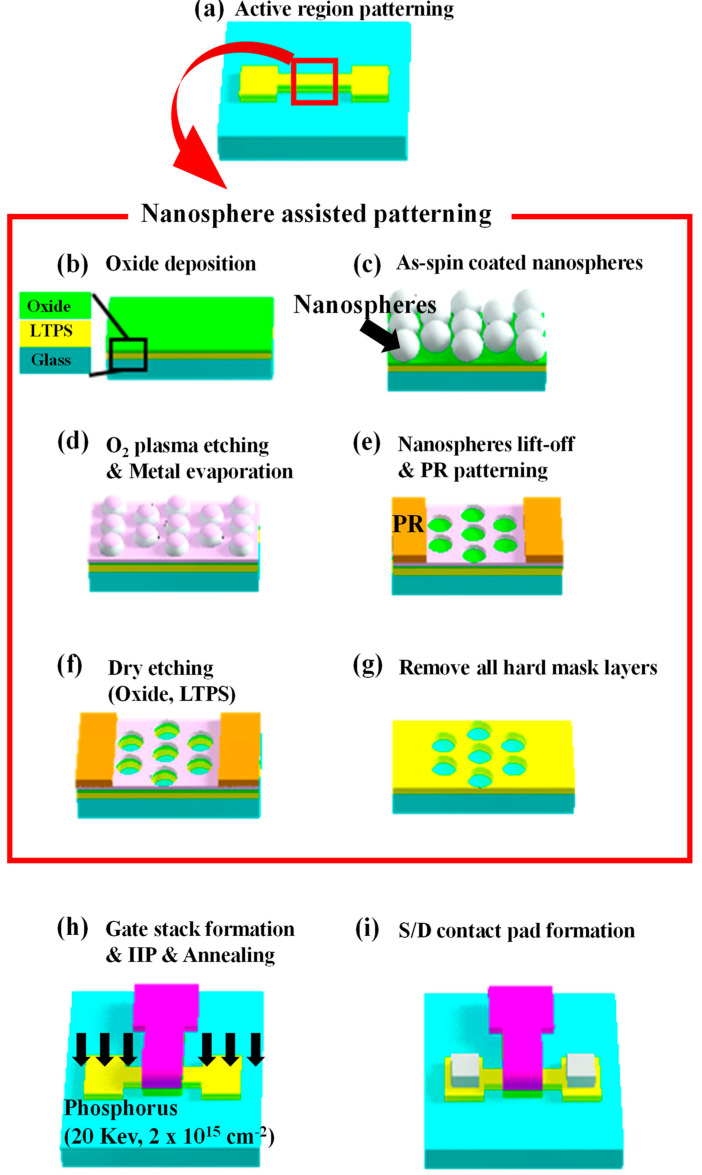
Schematic diagram of fabrication steps of nanonet-channel thin-film transistors (TFTs) using the nanosphere-assisted patterning (NAP) technique.

**Figure 2 micromachines-12-00741-f002:**
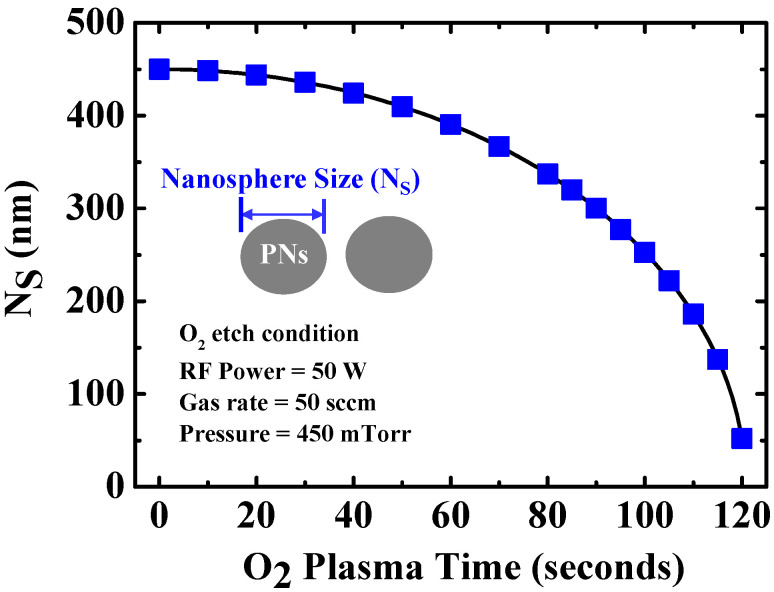
Size of the polystyrene nanosphere as a function of O_2_ plasma exposure time.

**Figure 3 micromachines-12-00741-f003:**
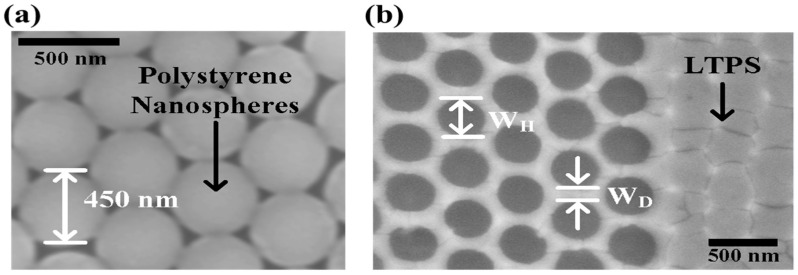
Top SEM images of (**a**) spin coated nanosphere after Figure 1c and (**b**) nanonet-channel after Figure 1g.

**Figure 4 micromachines-12-00741-f004:**
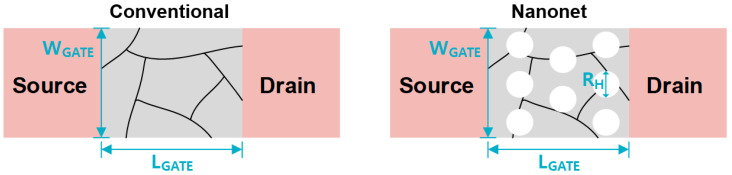
Top-view scheme of conventional planar TFTs (CON_TFTs) (**left**) and nanonet-channel low-temperature polysilicon TFTs (NET_TFTs) (**right**).

**Figure 5 micromachines-12-00741-f005:**
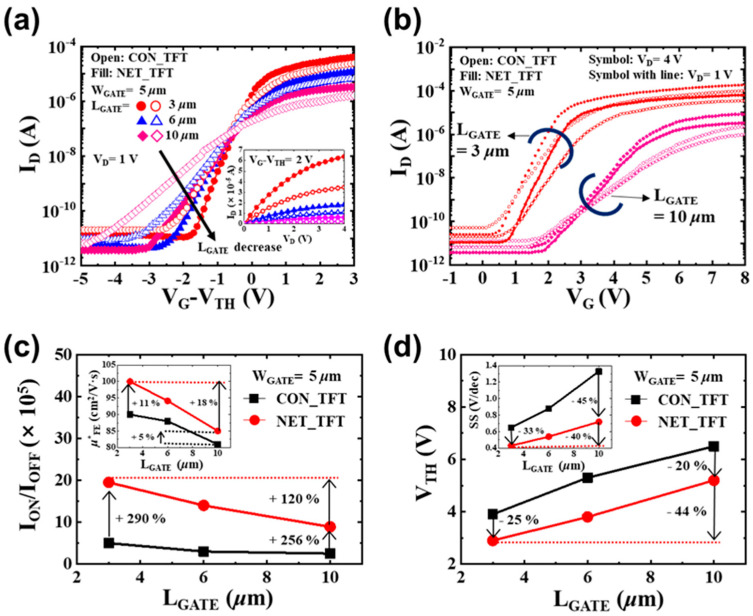
Comparison of the DC characteristics for NET_TFTs and CON_TFTs with different L_GATE_: (**a**) representative transfer curves (I_D_ vs. V_G_-V_TH_), (**b**) representative transfer curves with V_D_ = 1 and 4 V, (**c**) an average ON/OFF current ratio, and (**d**) an average V_TH_. Insets show the measured output curves (I_D_ vs. V_D_), *μ*^*^_FE_, and SS as a function of L_GATE_.

**Figure 6 micromachines-12-00741-f006:**
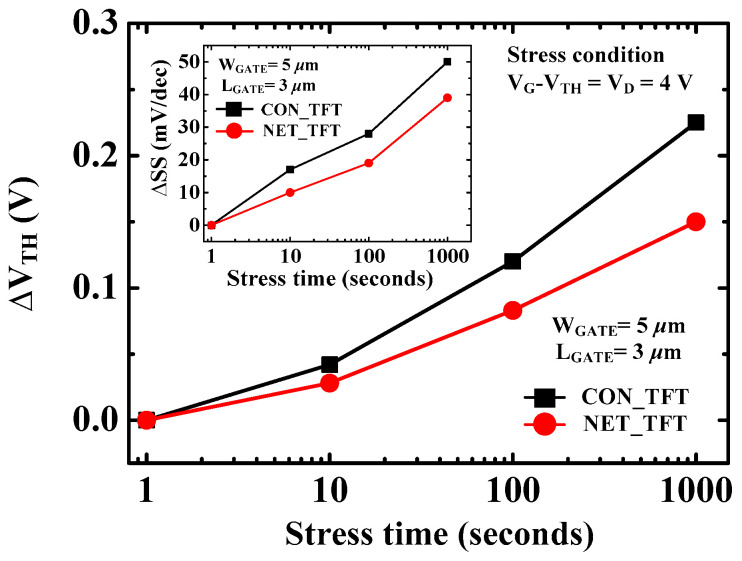
Measured ΔV_TH_ and ΔSS (inset) for NET_TFT and CON_TFT with L_GATE_ = 3 μm under an HCI stress condition at V_G_ − V_TH_ = V_D_ = 4 V during 0–1000 s.

**Figure 7 micromachines-12-00741-f007:**
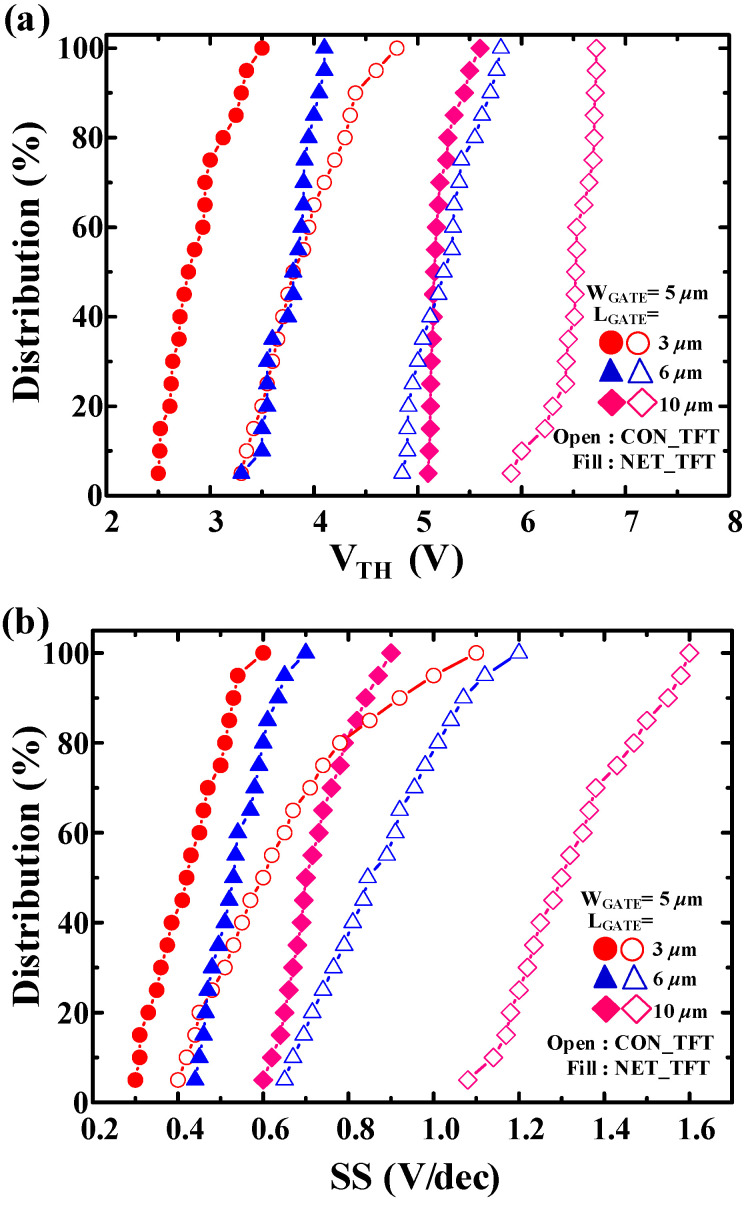
Cumulative distribution of (**a**) V_TH_ and (**b**) SS for NET_TFT and CON_TFT (20 devices).

**Figure 8 micromachines-12-00741-f008:**
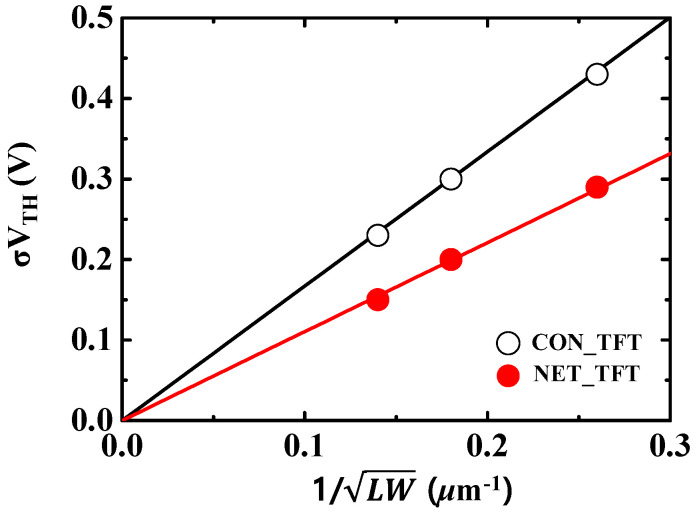
Standard deviation of V_TH_ (σV_TH_) as a function of *LW*^−^^1/2^ for both CON and NET_TFTs.

**Table 1 micromachines-12-00741-t001:** Average and standard deviation of V_TH_ and SS for NET_TFTs and CON_TFTs with different L_GATE_.

Channel Structure	L_GATE_ [μm]	AVG. V_TH_ [V]	STD V_TH_ [V]	AVG. SS [V/dec]	STD SS [V/dec]
conventional	3	3.9	0.43	0.65	0.20
	6	5.3	0.30	0.88	0.16
	10	6.5	0.23	1.33	0.15
nanonet	3	2.9	0.29	0.43	0.09
	6	3.8	0.20	0.54	0.08
	10	5.2	0.15	0.72	0.07

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
