# Peer review of "Fabrication and Characterization of Nanonet-Channel LTPS TFTs Using a Nanosphere-Assisted Patterning Technique"

_micromachines, 2021, doi:10.3390/mi12070741_

Round 1

Reviewer 1 Report

As one with only peripheral knowledge of current thin film technology, I would like the authors to provide more introductory information related to the process along the lines of the following:

A clear description of the mechanism that connects the nanonet to lower grain boundary traps -- How do the traps evolve in the process? What is the nature of grain boundaries in as deposited films? What process steps impact the grain boundaries? How does a mid-process step like the net forming have an impact? Is the improvement in grain boundaries just due to changes in stress due to hole formation or is it a result of realignment due to following anneals? 

In section 2, start with the LTPS deposition --it is confusing to assume the reader understands this structure. 

Data and analysis looks well written. 

Author Response

Dear Micromachines Editor

We would like to resubmit our paper titled "Fabrication and Characterization of Nanonet-Channel LTPS TFTs using a Nanosphere-Assisted Patterning Technique" for possible publication in Micromachines. 

We upload: (a) our point-by-point response to the comments (response to reviewers), (b) a revised manuscript with "Track Changes" indicating changes. 

We believe that our manuscript would be of interest to the broad audience of your journal and we would be grateful if the manuscript could be revaluated for publication in Micromachines.

yours Truly

Professor Jeong-Soo Lee

Reviewer 2 Report

The authors use NAP technique by using oxygen plasma method to implement LPTS TFTs. Even the results are interesting and useful, there are still some issues need authors to clarify before it can be accepted for publication.

  1. The two diagrams shown in Fig4a are not consistent. Please explain why? (For example: At Id-Vg, Id=8e-7(con.TFT@ Vg-Vt=2V, Vd=1V); The value is 2e-5 at Vd-Id one)
  2. In most cases, the leakage for long channel conventional TFTs should be lower than the shorters. Why are the leakages in your 10um con_TFT   higher than those in 6um devices?
  3. The SS trend for 10um con_TFTs seems not so reasonable. How many devices did you measure when you test it?
  4. The STD Vth for your devices is too large. Comparing it with your mean value, it means your porcess is very unstable. Why?

Author Response

Dear Editor,

We would like to resubmit our paper titled “Fabrication and Characterization of Nanonet-Channel LTPS TFTs using a Nanosphere-Assisted Patterning Technique” for possible publication in Micromachines.

We upload: (a) our point-by-point response to the comments (response to reviewers), (b) a revised manuscript with “Track Changes” indicating changes.

We believe that our manuscript would be of interest to the broad audience of your journal and we would be grateful if the manuscript could be revaluated for publication in Micromachines.

Yours Truly

Professor Jeong-Soo Lee

Round 2

Reviewer 2 Report

This version is much better than the previous one. Before it can be accepted, please clarify the description on Fig. 5.

1.Is the curve of Fig. 5 measured by a specific device or recalculated by the average data?

2. You only demonstrated ID vs (VGS-VTH)@VD=1V (linear region), please add one more plot @VD=4V(or higher at saturation region).

Author Response

Dear Editor,

We would like to resubmit our paper titled “Fabrication and Characterization of Nanonet-Channel LTPS TFTs using a Nanosphere-Assisted Patterning Technique” for possible publication in Micromachines.

According to reviewer#2’s comments, Fig. 5 is replotted including Id vs. Vg at Vd = 1 and 4 V, and the related paragraph is revised as well.

We upload: (a) our point-by-point response to the comments (below) (response to reviewers) and (b) a revised manuscript with “Track Changes” indicating changes.

Yours Truly

Professor Jeong-Soo Lee
